# Nuclear HKII-P-p53 (Ser15) Interaction is a Prognostic Biomarker for Chemoresponsiveness and Glycolytic Regulation in Epithelial Ovarian Cancer

**DOI:** 10.3390/cancers13143399

**Published:** 2021-07-07

**Authors:** Chae Young Han, David A. Patten, Se Ik Kim, Jung Jin Lim, David W. Chan, Michelle K. Y. Siu, Youngjin Han, Euridice Carmona, Robin J. Parks, Cheol Lee, Li-Jun Di, Zhen Lu, Karen K. L. Chan, Ja-Lok Ku, Elizabeth A. Macdonald, Barbara C. Vanderhyden, Anne-Marie Mes-Masson, Hextan Y. S. Ngan, Annie N. Y. Cheung, Yong Sang Song, Robert C. Bast, Mary-Ellen Harper, Benjamin K. Tsang

**Affiliations:** 1Departments of Obstetrics & Gynecology and Cellular & Molecular Medicine, Centre for Infection, Immunity and Inflammation, Interdisciplinary School of Health Sciences, University of Ottawa, Ottawa, ON K1N 6N5, Canada; 2Chronic Disease Program, Ottawa Hospital Research Institute, Ottawa, ON K1Y 4E9, Canada; 3Department of Experimental Therapeutics, The University of Texas MD Anderson Cancer Center, Houston, TX 77030, USA; 4Department of Biochemistry, Microbiology and Immunology, Faculty of Medicine, Ottawa Institute of Systems Biology, University of Ottawa, Ottawa, ON K1N 6N5, Canada; 5Department of Obstetrics and Gynecology and Cancer Research Institute, Seoul National University College of Medicine, Seoul 03080, Korea; 6Department of Obstetrics and Gynecology, LKS Faculty of Medicine, The University of Hong Kong, Hong Kong, China; 7Centre de Recherche du Centre Hospitalier de l’Université de Montréal and Institut du Cancer de Montréal, Montréal, QC H2X 0A9, Canada; 8Regenerative Medicine Program, Ottawa Hospital Research Institute, Ottawa, ON K1Y 4E9, Canada; 9Department of Pathology, Seoul National University College of Medicine, Seoul 03080, Korea; 10Faculty of Health Sciences, University of Macau, Macau 999078, China; 11Korean Cell Line Bank, Cancer Research Institute, Seoul National University College of Medicine, Seoul 03080, Korea; 12Cancer Therapeutics Program, Ottawa Hospital Research Institute, Ottawa, ON K1Y 4E9, Canada; 13Department of Pathology, LKS Faculty of Medicine, The University of Hong Kong, Queen Mary Hospital, Hong Kong, China

**Keywords:** epithelial ovarian cancer, chemoresistance, cancer metabolism, hexokinase II, P-p53 (Ser15)

## Abstract

**Simple Summary:**

Hexokinase II (HKII) is a key glycolysis enzyme associated with tumorigenesis, but its molecular mechanism and pathophysiological role in chemoresistant ovarian cancer remain elusive. In this study, we delineate the novel mechanism showing that activated phosphorylated-p53 (P-p53 Ser15) is required for the regulation of HKII intracellular trafficking and metabolic regulation in chemosensitive ovarian cancer, but not in chemoresistant ovarian cancer harboring p53 mutation. We have observed that increased nuclear HKII-P-p53 (Ser15) interaction is likely associated with chemosensitivity and better survival outcomes in epithelial ovarian cell lines, human primary epithelial ovarian cancer cells, and tumor sections. Nuclear HKII-P-p53 (Ser15) interaction may function as a promising prognostic biomarker, enabling prediction of patients with poor prognosis for deciding better clinical strategies.

**Abstract:**

In epithelial ovarian cancer (EOC), carboplatin/cisplatin-induced chemoresistance is a major hurdle to successful treatment. Aerobic glycolysis is a common characteristic of cancer. However, the role of glycolytic metabolism in chemoresistance and its impact on clinical outcomes in EOC are not clear. Here, we show a functional interaction between the key glycolytic enzyme hexokinase II (HKII) and activated P-p53 (Ser15) in the regulation of bioenergetics and chemosensitivity. Using translational approaches with proximity ligation assessment in cancer cells and human EOC tumor sections, we showed that nuclear HKII-P-p53 (Ser15) interaction is increased after chemotherapy, and functions as a determinant of chemoresponsiveness as a prognostic biomarker. We also demonstrated that p53 is required for the intracellular nuclear HKII trafficking in the control of glycolysis in EOC, associated with chemosensitivity. Mechanistically, cisplatin-induced P-p53 (Ser15) recruits HKII and apoptosis-inducing factor (AIF) in chemosensitive EOC cells, enabling their translocation from the mitochondria to the nucleus, eliciting AIF-induced apoptosis. Conversely, in p53-defective chemoresistant EOC cells, HKII and AIF are strongly bound in the mitochondria and, therefore, apoptosis is suppressed. Collectively, our findings implicate nuclear HKII-P-p53(Ser15) interaction in chemosensitivity and could provide an effective clinical strategy as a promising biomarker during platinum-based therapy.

## 1. Introduction

Carboplatin—a standard, first-line, platinum-based chemotherapy for EOC—is an analogue of cisplatin (CDDP), which is often used in case of relapse [1,2]. However, 70% of EOC patients develop chemoresistance within 15 months, leading to poor 5-year overall survival (OS) rates (30–50%) [1]. We and others have demonstrated that the mechanism of chemoresistance is multifactorial, largely due to a defect in key tumor suppressors (e.g., p53) or activation of oncogenes (e.g., *Akt*, *PI3K*) [3,4]. 

*TP53* is frequently mutated in EOC (>70%), especially in high-grade serous (HGS) histotypes—the most prevalent subtype (>90%) [5,6], often associated with chemoresistance. p53 encoded by *TP53* is a key regulator of apoptosis, and is rapidly increased in response to DNA-damaging agents, such as CDDP [7]. Activation/stabilization of p53 occurs through its site-specific phosphorylation at Ser15 and Ser20 in response to CDDP [4]. p53 also governs multiple cellular processes, including DNA repair, cell cycle, apoptosis, and cell metabolism [8,9]. We previously reported that p53 is critical for the induction of apoptosis in chemoresistant cells, even when metabolism is suppressed [10]. 

Elevated aerobic glycolysis, often known as the Warburg effect, is a common characteristic of cancer [11]. The metabolic shift from oxidative phosphorylation (OXPHOS) to glycolysis in cancer cells is mediated by the dysregulation of glycolytic enzymes [12]. Metabolic enzymes are also regulated by the interplay between tumor suppressors and oncogenes [13,14]. Hexokinase II (HKII) is a key enzyme that catalyzes the first step in glycolysis, converting glucose to glucose-6-phosphate. Among five isoforms of HK (HKI–HKV), HKII is highly associated with cancer cell survival and tumorigenesis [13,15,16].

HKII is mainly localized in the mitochondria (80%) [17], and binds to voltage-dependent anion channels (VDAC), maintaining mitochondrial membrane integrity by suppressing the release of apoptotic molecules (e.g., cytochrome c and apoptosis-inducing factor (AIF)) [18,19]. HKII bound to mitochondria (mito-HKII) enhances the tight coupling of glucose phosphorylation for ATP generation [20,21]. DeWaal et al. demonstrated that forced expression of mitochondrial binding-deficient HKII failed to restore cancer cell proliferation and tumorigenesis, suggesting a pivotal role of mito-HKII [22]. Phosphorylated Akt promotes the binding of HKII to the outer mitochondrial membrane (OMM) [23]. Mito-HKII protects cells against Ca^2+^ overload and the opening of mitochondrial permeability transition pores (mPTP), thus preventing cell rupture [24]. Whether or not the detachment of mito-HKII and impaired glycolysis influences chemosensitivity is unclear.

AIF is a mitochondrial intermembrane flavoprotein [25]. In response to cell death signaling, AIF is released from the mitochondria and translocated to the nucleus, inducing chromatin condensation in a caspase-independent manner [26]. We have previously reported that the activation of Akt attenuates AIF-induced apoptosis and nuclear AIF accumulation [27].

In the present study, we examined whether the clinicopathological expression patterns and cellular localization of HKII are associated with CDDP chemosensitivity in EOC. Here, we assessed whether nuclear HKII-P-p53 (Ser15) interaction is correlated with chemosensitivity and functions as prognostic biomarker in paired chemosensitive and chemoresistant EOC cell lines, human primary EOC cells, and ovarian tumor sections, using a proximity ligation assay (PLA). 

Mechanistically, we tested the following hypothesis (graphical abstract). In chemosensitive cells, treatment with CDDP increases P-p53 (Ser15), P-p53 (Ser15) facilitates the translocation of HKII–AIF from the mitochondria to the nucleus, and nuclear AIF induces apoptosis. In chemoresistant cells, treatment with CDDP does not affect HKII and AIF strongly bound in the mitochondria, maintaining HKII enzymatic activity. As a result, P-p53 (Ser15)-induced HKII suppression and AIF-induced apoptosis are prevented. 

## 2. Materials and Methods

Cell lines and culture: EOC cell lines of different histological subtypes (HGS (OVCAR-3 and OV- 2295), endometrioid (A2780 and A2780cp) [28], and clear cell (CC and ES-2)) were provided by Dr. Rakesh Goel, Dr. Barbara Vanderhyden (Ottawa Hospital Research Institute, OHRI, Ottawa, ON, Canada), and Dr. Anne-Marie Mes-Masson (CHURM). The parental sensitive A2780s cells were treated with CDDP with every 2–3 passages in order to develop their cisplatin-resistant counterpart, A2780cp. Cell lines were cultured at 37 °C with 5% CO_2_ in RPMI 1640 medium supplemented with 10% heat-inactivated fetal bovine serum (FBS), penicillin/streptomycin (10,000 U/mL), and amphotericin B (0.5 μg/mL). Detailed information is provided in Appendix A [5,27,29,30]. 

Primary human EOC cells: Primary human EOC cell cultures were obtained from Dr. Barbara Vanderhyden (Ottawa Ovarian Cancer Tissue Bank) and Ja-Lok Ku (Korean Cell Line Bank, Seoul National University Hospital, SNU). Tumor ascites and primary ovarian tumor cell cultures/ascites were collected from EOC patients with subtypes of HGS and CC using Institutional Review Board (IRB)-approved protocols (SNU: 1409-1540-616 and OHSN-REB 1999540-01H) with patient informed consent. Ascites-derived EOC cells were maintained in either DMEM or RPMI 1640 supplemented with 10% FBS and penicillin/streptomycin (10,000 U/mL); the passage numbers of these cell lines were less than 10. Detailed information is provided in Appendix A.

Immunohistochemistry (IHC) sections: Under IRB-approved protocols at the Ottawa Health Science Network Research Ethics Board (OHSN-REB Protocol No.:20150646-01H) and collaborating institutions—including the Seoul National University Hospital (IRB No.:H-1711-142-904), University of Hong Kong (IRB UW16_107), and CRCHUM (IEC No. 2005-1893, BD 04.002-BSP)—formalin-fixed paraffin-embedded (FFPE) ovarian tumor sections were collected and assessed. The stage, histology, and grade of cancer were determined using International Federation of Gynecology and Obstetrics (FIGO) criteria. Pre-chemotherapy and post-chemotherapy ovarian tumor sections were obtained at primary and secondary cytoreductive surgery, respectively, in case of relapse, except for one neoadjuvant case. Detailed information is provided in Appendix A.

Proximity ligation assays: Proximity ligation assays (PLAs) were conducted using the Duolink Detection Kit (Sigma, Ronkonkoma, NY, USA), as previously described [31,32]. Briefly, 8-well change slides were incubated overnight with a primary pair of either HKII and P-p53 (Ser15) or HKII and AIF, as indicated (Appendix A). Slides were then incubated with secondary proximity probes (PLA probe PLUS and MINUS), solution containing ligase and amplification polymerase, and a fluorophore with 594 nm excitation and 624 nm emission. Cells were then counterstained with TOM 20 (mitochondrial marker) or DAPI (nucleus marker). Images were obtained via confocal microscopy (LSM 510, Carl Zeiss, 64X objective), and positive signals were analyzed using the Duolink Image Tool program (Sigma). At least 50 cells were counted for each group. For PLAs in ovarian tumor sections, 4–5 micron sections were heated in citrate buffer for antigen retrieval (20 min) and incubated with primary antibodies. The median value (*n* = 41) was assigned as the cutoff value (HKII-P-p53 (Ser15)) to define low PLA expression with a score of <0.8 and high PLA expression with a score of ≥0.8. Kaplan–Meier analysis was used to plot survival curves using the logrank method. 

Apoptosis: Apoptosis was morphologically assessed using Hoechst nuclear staining, as previously described [33]. At least 400 cells were counted for each group, and the process was blinded to avoid experimental bias. 

Western blot (WB): Protein extraction and Western blot (WB) analysis were performed as previously described [27]. Unless otherwise indicated, blots were incubated overnight at 4 °C with primary antibodies (Appendix A). The membrane was then washed with TTBS and incubated with fluorescence-conjugated goat-anti-rabbit or anti-mouse secondary antibodies, followed by quantification and analysis using LI-COR (Odyssey Imager, Lincoln, NE, USA). Alternatively, membranes were incubated with horseradish peroxidase (HRP)-conjugated secondary goat-anti-rabbit antibodies, and visualized using a Supersignal Chemiluminescence Kit (Thermofisher Scientific, Waltham, MA, USA). 

Immunoprecipitation (IP): Protein lysates (1 mg) were incubated at 4 °C overnight with rabbit monoclonal anti-P-p53 (Ser15) antibody (1 µg, #9284, Cell Signaling, Danvers, MA, USA), or rabbit IgG (1 µg, sc-3888, Santa Cruz, CA, USA) as a control, and immunoprecipitated (RT, 2 h) with 30 µL Protein A/G Beads (sc-2003, Santa Cruz, CA, USA), as previously described [33]. Beads were pelleted, washed, and resuspended in sample buffer, before being boiled and loaded onto 8% SDS-PAGE. Proteins were then transferred to nitrocellulose membranes, and HKII and P-p53 (Ser15) were examined via Western blot using Rabbit TrueBlot Secondary Antibody (Rockland, PA, USA). 

Immunofluorescence: Immunofluorescence microscopy was performed as previously described [34]. Cells were seeded in 8-chamber slides (BD Biosciences) 1 day before analysis and treated as indicated. Cells were fixed with paraformaldehyde (4% at 4 °C overnight), washed with PBS, incubated with Triton X-100 (0.5% for 5–10 min), blocked with BSA (3%), and incubated with primary antibodies (Appendix A). The slides were then incubated with fluorescence-conjugated secondary antibodies (Alexa Fluor 488/555) and counterstained with DAPI. Images were obtained on a Zeiss LSM 510 (Carl Zeiss, Germany) inverted confocal scanning microscope or a ZEISS AXIO Observer immunofluorescence microscope, with appropriate argon lasers (488 and 546 nm, respectively), and 40–64x objective. At least 100 cells were analyzed per experimental group. 

Gene transduction by adenoviral infection: Cells were infected with adenoviral constructs containing wild-type p53, LacZ as control, (multiplicity of infection, MOI = 0–1.0, 12 h), and dominant-negative (DN)-Akt with triple mutation (MOI = 40, 12 h). DN-Akt—a dead kinase mutant with a triple phosphorylation site mutated with alanine (K179A, T308A, and S473A)—was used to suppress the function of endogenous Akt. 

Overexpression: HKII cDNA (RC209482) and control vectors (PS100001) were purchased from Origene (Rockville, MD, USA), and then amplified and purified using a DNA Miniprep Kit (Qiagen, MD, USA). Mutation of the nuclear localization sequence (NLS) (KRFRK → ERFRD, KRLHK → ERLHE) in an HKII cDNA vector was carried out using Vector Builder (Chicago, IL, USA). A2780s cells were transfected using Lipofectamine 3000 with the indicated vectors, treated with CDDP (10 μm) or vehicle (DMSO), and fixed for immunofluorescence studies or harvested for Western blot analyses. 

Extracellular flux assays (Seahorse): Metabolic measurements of the oxygen consumption rate (OCR) and the extracellular acidification rate (ECAR) were performed on EOC cell lines using the Seahorse XF96e Extracellular Flux Analyzer (Agilent, Santa Clara, CA, USA), as previously described [35]. Briefly, 20,000 cells/well were seeded on XF96e cell culture microplates 1 day before each experiment. Prior to the experiment, the culture medium was replaced with either XF96e DMEM or glucose-free DMEM for ECAR and incubated in a non-CO_2_ incubator (37 °C, 1 h). The mitochondrial stress test for the assessment of OCR (resting respiration, ATP-linked OCR, and maximal respiratory capacity) was performed following the sequential addition of oligomycin A (1 μM), carbonyl cyanide-p-trifluoromethoxyphenylhydrazone (FCCP, 0.5 μM), and antimycin A (0.5 μM)/rotenone (1 μM).

A glycolytic stress test for the assessment of ECAR (basal glycolysis, glycolytic capacity, and glycolytic reserve) was performed following the sequential addition of glucose (10 mM), oligomycin (1.0 μM), and 2-deoxy-D-glucose (DG) (50 mM) in an XF96e flux analyzer. Both OCR and ECAR were measured over a 3-min period, and these values were normalized to the protein concentration, as determined by the Bradford assay.

Statistical analysis: Results are expressed as the mean ± SEM of at least three independent experiments. Statistical analysis was performed via one-way or two-way ANOVA, using Prism (version 7.0; Graph Pad, San Diego, CA, USA). Correlations were analyzed by the Pearson method. Differences between multiple groups were determined by the Bonferroni post-hoc test. Statistical significance was inferred at *p* < 0.05.

## 3. Results

### 3.1. Increased Nuclear HKII-P-p53 Interaction as a Potential Biomarker in EOC

Progression-free interval (PFI)—the duration from the termination of chemotherapy to relapse [2]—is frequently used as an indicator of chemoresponsiveness, and is generally defined by a 6-month (m) period: chemosensitivity (PFI > 6 m) and chemoresistance (PFI ≤ 6 m). Using this standard for chemoresponsiveness, we first performed a PLA assessment in EOC tumor sections (*n* = 41 pairs) of different histological subtypes (36 HGS, 4 clear cell, and 1 endometrioid) obtained pre- and post-chemotherapy from the same EOC patients (Figure 1A,B), in order to determine whether nuclear localization of HKII-P-p53 (Ser15) interaction is associated with prognosis and chemoresponsiveness as determined by the length of progression-free interval [36,37] (Appendix A). When comparing the nuclear HKII-P-p53 (Ser15) interaction in post-chemotherapy relative to pre-chemotherapy sections, we observed a notable increase in nuclear HKII-P-p53 (Ser15) interaction in a chemosensitive cancer (PFI = 40 m) associated with longer PFI, but not in a chemoresistant cancer associated with shorter PFI (PFI = 1 m) (Figure 1C). An increased nuclear HKII-P-p53 (Ser15) interaction was expressed as:

HKII-P-p53 (Ser15) _(post-chemotherapy)_–HKII-P-p53 (Ser15) _(pre-chemotherapy)_ and was positively correlated with PFI (Figure 1D; r = 0.72; **** *p* < 0.0001) and OS (Figure 1E; r = 0.43; ** *p* = 0.004). The boxplot also showed that the increased PLA value of nuclear HKII-P-p53 (Ser15) interaction was significantly elevated in samples collected from intermediate chemosensitive patients with prolonged PFIs at 6 m < PFI ≤ 12m (* *p* < 0.05) and 12 m < PFI (**** *p* < 0.0001), compared with samples collected from chemoresistant patients (PFI ≤ 6m) (Figure 1F). Finally, Kaplan–Meier analysis indicated that patients (*n* = 41) with a higher nuclear HKII-P-p53 (Ser15) interaction after chemotherapy had significantly prolonged PFIs (Figure 1G; ****p* = 0.0006), PFS (Figure 1H; **** *p* < 0.0001), and OS (Figure 1I; *** p* = 0.008) compared to patients with lower nuclear HKII-P-p53 (Ser15) interaction. We also computed receiver operating characteristic (ROC) curves in order to determine whether the nuclear HKII-P-p53 (Ser15) interaction could function as a prognostic biomarker predicting chemosensitivity. For PFI, nuclear HKII-P-p53 (Ser15)-positive cases showed an area under the curve (AUC: 0.844; Appendix A) > 0.5 and showed a 95% confidence interval (CI) of 0.690–0.990. For OS, nuclear HKII-P-p53 (Ser15) interaction showed an AUC of 0.708 (95% CI: 0.552–0.963; Appendix A). ROC curves for PFI showed AUC values > 0.8, suggesting that nuclear HKII-P-p53 (Ser15) interaction has strong prognostic value for chemoresponsiveness. However, the ROC curve for OS is less robust, showing lower AUC values of < 0.8. Taken together, our findings suggest that nuclear HKII-P-p53 (Ser15) interaction has better prognostic value for chemoresponsiveness than for survival outcomes. Collectively, these data suggest that increased nuclear HKII-P-p53 (Ser15) interaction after chemotherapy is more likely associated with chemoresponsiveness and prolonged OS in EOC.

### 3.2. CDDP Increases Nuclear HKII-P-p53 (Ser15) Interaction in Chemosensitive Primary Human Ovarian Cultures 

Considering that heavily passaged cell lines do not always reflect the biological phenotype of the tumors from which they were derived [38], we extended the above studies to primary human ovarian/ascites cultures from EOC patients with different PFIs. After CDDP treatment (10 µM; 24 h), the nuclear intracellular trafficking and interaction of HKII-P-p53 (Ser15) (as PLA unit: red signal) were highly increased in chemosensitive SNU-3561 cells (PFI = 19 m), but this response was significantly attenuated in chemoresistant 2068 cells (PFI = 0 m) (Figure 2A).

Next, we observed significantly lower nuclear HKII-P-p53 (Ser15) interaction in chemoresistant primary human EOC cells (PFI ≤ 6 m), compared with that of chemosensitive SNU-3561 cells (Figure 2B; ***** p* < 0.0001). Then, we determined whether increased nuclear interaction of HKII-P-p53 (Ser15) was correlated with the PFIs of the patients from whom the cells were isolated. The results shown in Figure 2C clearly demonstrate that PFI was positively correlated with nuclear HKII-P-p53 (Ser15) interaction (Figure 2C; r = 0.73; * *p* = 0.011).

When we assessed the nuclear HKII-P-p53 (Ser15) interaction in the context of the chemosensitivity of primary EOC cells (CDDP-induced apoptotic rate), using Pearson’s correlation analysis, we observed a positive correlation (Figure 2D; r = 0.88; *** *p* = 0.0007). In addition, CDDP-induced chemosensitivity of primary EOC cells (apoptotic rate) was correlated with PFI (Figure 2E; r = 0.68; * *p* = 0.03). In summary, nuclear HKII-P-p53 (Ser15) interaction occurred in primary EOC human cells from chemosensitive (PFI > 6 m) patients, but not in chemoresistant (PFI ≤ 6 m) patients, suggesting that increased nuclear intracellular trafficking of HKII-P-p53 (Ser15) is a determinant of chemosensitivity.

### 3.3. Mito-HKII is Associated with Chemoresistance and Glycolysis in EOC

Mito-HKII is important for cell survival during cell stress [18,23], but the manner in which mito-HKII contributes to chemoresistance remains unclear. Three monopartite (single stretches of basic amino acids) nuclear localization signals (NLSs: K-K/R-X-K/R) exist in the amino acid sequence of HKII (Figure 3A), showing that HKII can be transported to the nucleus. 

We first examined the apoptotic rate in multiple subtypes of EOC cells, including HGS (OV-2295, OVCAR-3), endometrioid (paired A2780s and A2780cp), and CC (ES-2) EOC cells. CDDP significantly increased the apoptotic rate up to 20% in OV-2295 and A2780s, but not in chemoresistant OVCAR-3, A2780cp, and ES-2 cells (Figure 3B ***** p* < 0.0001). We then examined whether CDDP (10 μM; 0–24 h) alters the subcellular localization of HKII in EOC. Strong mito-HKII localization was observed in chemoresistant HGS (OVCAR-3) and clear cell (ES-2) subtypes of EOC cells (Figure 3C), suggesting that mito-HKII is associated with chemoresistance, and that this phenomenon is not specific to the HGS subtype. This notion is supported by the observation that nuclear HKII localization in chemosensitive endometrioid p53 wild-type A2780s cells was significantly increased—from 28% to 55%—by 24 h after CDDP treatment (Figure 3D; *** p* < 0.01), whereas only 17% of HKII was observed in the nuclear region of the p53-mutant chemoresistant counterpart (A2780cp) and was unaffected by CDDP. Conversely, mito-HKII was significantly decreased—from 60% to 35%—at 24 h in chemosensitive cells (Figure 3D; * *p* < 0.05), whereas HKII was predominantly localized in the mitochondria (70%) in the chemoresistant cells, and unaffected by CDDP. In primary human EOC cells, CDDP also significantly induced HKII nuclear localization in chemosensitive A-8 cells (HGS; PFI = 39 m; ** p* < 0.05; Appendix A), but not in chemoresistant A-39 cells (CC; PFI = 2 m). Notably, strong HKII nuclear localization was observed in chemosensitive SNU-3561 (HGS; PFI = 16 m), even without CDDP treatment, suggesting that low levels of basal nuclear HKII could be a determinant of chemosensitivity (Appendix A), showing high correlation between basal nuclear HKII and apoptosis (r = 0.81; *p* = 0.03). 

Nuclear localization of P-p53 (Ser15) is required for apoptotic function and transcriptional regulation [9,39]. We examined whether CDDP facilitates the colocalization of P-p53 (Ser15) and HKII in the nucleus. In chemosensitive OV-2295 HGS cancer cells, we observed significant nuclear colocalization of HKII and P-p53 (Ser15), starting at 6 h (Appendix A; *** *p* < 0.001), and reaching a maximum at 24 h (***** p* < 0.0001), but nuclear localization was not increased in chemoresistant OVCAR-3 (HGS). Similarly, we observed strong HKII-P-p53 (Ser15) colocalization in chemosensitive endometrioid A2780s cells (Appendix A; ** *p* < 0.01), but not in isogenic chemoresistant A2780cp cells. However, we observed significantly increased sole nuclear localization of P-p53 (Ser15) with CDDP treatment, regardless of chemosensitivity (Appendix A). 

Elevated aerobic glycolysis is a common characteristic of cancer cells. We investigated whether chemotherapy and HKII detachment from mitochondria affected subsequent energy metabolism [18,23]. Paired isogenic chemosensitive and chemoresistant cancer cell lines—A2780s and A2780cp, respectively—were cultured with different concentrations of CDDP (0–10 μM; 24 h), followed by measurement of glycolysis as an extracellular acidification rate (ECAR) via extracellular flux analysis (Seahorse). CDDP markedly decreased the levels of resting ECAR (glycolysis; Figure 3E; *** p* < 0.01) and the rate of both resting and maximal ECAR (Figure 3F; ** p* < 0.05) in chemosensitive (A2780s) cells, but not in chemoresistant (A2780cp) cells. Notably, chemoresistant A2780cp cells showed higher resting and maximal ECAR rates than did A2780s cells, implying ECAR’s high dependency on glycolysis. CDDP significantly decreased the glucose consumption of chemosensitive cells, starting from 5 μM, and reaching a maximum decrease (~50%) at 10 μM (Figure 3G; *** *p* < 0.001), but had no effect on A2780cp cells. Similarly, CDDP (10 μM) significantly decreased HK activity in A2780s cells, but not in A2780cp cells (Figure 3H; ** *p* < 0.01). HKII knockdown in A2780cp showed a significant decrease of ECAR and glucose consumption (Appendix A), although the response was relatively small. Collectively, this suggests that detachment of HKII induced by CDDP decreased glycolysis.

### 3.4. p53 and HKII Nuclear Localization Signal Together Are Required for Intracellular HKII Trafficking

We then determined whether the detachment of mito-HKII affects its metabolic function and aerobic glycolysis, and whether activated P-p53 (Ser15) can facilitate nuclear HKII localization. HKII nuclear localization was increased twofold (from 20% to 40%; Figure 4A; *** p* < 0.01) in A2780cp cells infected with an adenoviral construct (Adv) of wild-type p53 (multiplicity of infection (MOI) = 1.0; 12 h) and treated with CDDP (10 μM; 24 h) compared with Adv–LacZ as control (CTL) alone. In these cells, nuclear colocalization of P-p53 (Ser15) and HKII was significantly increased, from 23% to 45% (Figure 4A; ** *p* < 0.01), but this phenomenon was attenuated in A2780cp cells infected with Adv–LacZ only, possibly due to defective endogenous p53. Notably, neither Adv–p53 alone nor Adv–LacZ (as counterparts without CDDP) elicited such responses, suggesting that CDDP-activated P-p53 (Ser15) is required for modulating HKII cellular localization.

Glycolysis was decreased in A2780cp cells infected with Adv–p53 and treated with CDDP (10 μM; 24 h) (Figure 4B; ** *p* < 0.01). The p53 infection itself significantly decreased glycolysis, suggesting that both p53 itself and P-p53 (Ser15) enforce this metabolic regulation (Figure 4B; ## *p* < 0.01). Similarly, HK activity was significantly decreased (Figure 4C; ** p* < 0.05) in p53-infected cells treated with CDDP, compared with LacZ as CTL. 

We previously reported that p53 is required for the induction of apoptosis in HKII-knockdown chemoresistant cells [10]. Here, we assessed whether the pharmacological inhibitor 3-bromopyruvic acid (3-BP)—shown to dissociate HKII from mitochondria treated together with CDDP [19]—sensitizes chemoresistant ovarian cancer cells. Consistent with our previous observations, high concentrations of 3-BP (50–100 µM, 6 h pre-treatment) followed by CDDP treatment (10 µM, 24 h) significantly increased apoptosis in Hey cells harboring p53 WT (Figure 4D), but not in chemoresistant OVCAR3 with mutant p53 (Figure 4E). Collectively, these results may indicate that p53 is required for facilitating the detachment of mito-HKII, and attributable to the dysregulation of HK enzyme activity and glycolysis associated with apoptotic response.

We also examined how the mutant HKII’s nuclear localization sequence (NLS) (KRFRK to EFRFD, KRLHK to ERLHE) affects HKII nuclear localization even in the presence of p53 WT, as confirmed by mutated vector construction and successful transfection (Appendix A). We observed that mutation of HKII’s NLS significantly decreased HKII nuclear localization—by half—but did not attenuate it completely (Figure 4F; ** *p* < 0.01); the decrease was 22% with the mutant vector and 45% with either the HKII WT vector or the control vector. Mutation of HKII also partly decreased the rate of apoptosis, from 27% to 15% (Figure 4G). These findings suggest that HKII’s NLS is necessary, but not sufficient, for nuclear localization and apoptotic signaling.

### 3.5. p53 Reconstitution and Akt Depletion Together Facilitate Detachment of HKII from the Mitochondria

Activated Akt has been reported to promote the binding of HKII to the OMM of mitochondria [23]. We then examined whether inhibition of Akt function in Akt-expressing ovarian cancer cells (A2780cp) with dominant negative Akt adenovirus (Adv–DN-Akt—a dead kinase mutated at three phosphorylation sites: K179A, T308A, and S473A) and p53–Adv reconstitution together would detach mito-HKII from the OMM and elicit chemosensitivity. A2780cp (p53-mutant) cells were infected with Adv–LacZ as a control (MOI = 0.5; 12 h), Adv–p53–GFP (MOI = 0.5; 12 h), or Adv–DN-Akt (MOI = 40; 12 h)—alone or together—followed by CDDP treatment (10 µM; 24 h). We observed that CDDP significantly decreased mito-HKII from 43% to 27% (** *p* < 0.01 vs. CTL, Figure 5A) in A2780cp cells co-infected with Adv–p53 and Adv–DN-Akt. Compared with CTL, nuclear HKII was significantly increased, threefold (**** *p* < 0.0001, Figure 5A), in A2780cp cells co-infected with p53 and DN-Akt followed by CDDP treatment.

In Figure 5B, Western blot (WB) was performed using the same cells as those treated above, and successful infection was confirmed as Akt and p53 overexpression. Interestingly, WB analysis showed that the protein content of HKII was not decreased in A2780cp cells infected with either functional p53 (Adv–p53) or DN-Akt alone, regardless of CDDP treatment. However, DN-Akt and p53 together decreased the protein content of HKII and p-Akt (347), irrespective of CDDP treatment. For the apoptotic rate, infection of p53 or DN-Akt alone did not show a significant increase for A2780cp cells infected with LacZ or p53. However, both Adv–p53 and Adv–DN-Akt together showed an increase of 20% in the apoptotic rate following CDDP treatment (*** p* < 0.01, Figure 5C). In Figure 5D,E, A2780cp cells infected with Adv–p53 and Adv–DN-Akt together synergistically decreased the metabolic activity of cells (ECAR and OCR, * *p* < 0.05) in a significant manner. Collectively, these data suggest that functional p53 and Akt depletion co-operatively facilitate the nuclear localization of HKII in chemoresistant EOC cells, due to metabolic regulation and induction of apoptosis.

### 3.6. CDDP Promotes Nuclear HKII-P-p53 (Ser15)–AIF interaction In Vitro

We previously reported that CDDP promotes the nuclear localization of AIF—a mediator of caspase-independent apoptosis—and others have observed that HKII interacts with AIF in the mitochondria [19,27,40]. We hypothesized that CDDP-induced P-p53 (Ser15) promotes the recruitment of HKII and AIF in chemosensitive EOC cells, enabling the translocation of this complex to the nucleus and eliciting AIF-induced apoptosis. To test this hypothesis, we examined the interaction between HKII-P-p53 (Ser15) and HKII–AIF, both by PLA (Figure 6A–D) and by immunoprecipitation (IP) (Figure 6E). In chemosensitive HGS cells (OV-2295), CDDP treatment (10 µM; 0–24 h) significantly promoted both mitochondrial (** p* < 0.05) and nuclear HKII-P-p53 (Ser15) interaction at 24 h (Figure 6B, *** p* < 0.01). Conversely, in chemoresistant HGS cells (OVCAR-3), PLA signaling was markedly increased in the mitochondrial region (Figure 6B, ***** p* < 0.0001) in response to CDDP. Similarly, in chemosensitive endometrioid cells (A2780s) treated with CDDP (10 µM; 0–24 h), nuclear HKII-P-p53 (Ser15) interaction was increased starting at 6 h and reached the highest levels at 24 h (Figure 6C; *** p* < 0.01)—but not in chemoresistant cells. Nuclear HKII–AIF interaction was notably increased after 24 h of CDDP treatment (Figure 6D; ** p* < 0.05), but not in chemoresistant A2780cp cells.

Interestingly, this interaction in the mitochondria was largely attenuated at 3–6 h, but recovered at 24 h, suggesting intracellular trafficking of HKII-P-p53 (Ser15) from the mitochondria to the nucleus. In contrast, chemoresistant endometrioid cells (A2780cp) exhibited minimal nuclear HKII-P-p53 (Ser15) interaction and were enriched in the mitochondrial region at 24 h (Figure 6C; ** p* < 0.05), supporting the suppression of the nuclear translocation of HKII-P-p53 (Ser15) interaction in chemoresistant EOC cells. On the other hand, HKII–AIF interaction in A2780s cells was increased in the nucleus (Figure 6D; ** p* < 0.05), but decreased in the mitochondria following CDDP treatment (24 h). Compared to A2780s cells, HKII–AIF interaction in chemoresistant A2780cp cells was weaker overall, being mainly localized in the mitochondria, irrespective of the presence of CDDP.

Consistent with previous findings with PLA, IP experiments with nuclear fractions (Figure 6E) demonstrated that that P-p53 (Ser15) notably binds to HKII (**** p* < 0.001) and AIF (Figure 6E; **** p* < 0.001) in chemosensitive A2780s cells, but not in chemoresistant A2780cp cells, in response to CDDP. Together, these results support the hypothesis that CDDP-induced P-p53 (Ser15) is a leading regulatory molecule that facilitates the translocation of HKII–AIF from the mitochondria to the nucleus, suggesting an interaction and co-movement of this complex (P-p53 (Ser15)–HKII–AIF) and eliciting AIF-mediated apoptosis.

## 4. Discussion

The molecular mechanism(s) of chemoresistance are multifactorial and are not completely understood. In the present study, we have demonstrated in HGS and other histologic subtypes of EOC that chemosensitivity is associated with dislocation of HKII from the mitochondria, and this is facilitated by wild-type-activated p53, with which HKII forms a complex and translocates to the nucleus upon platinum challenge. By contrast, in the presence of a mutated/deleted *TP53* gene in EOC, HKII remained within the mitochondria, leading to elevated glycolysis, cell survival, and chemoresistance. These studies provide novel mechanistic insights into the molecular signaling cascade in the regulation of metabolism and chemosensitivity in EOC.

Recent studies have shown that, upon platinum challenge, this tumor suppressor traverses to the mitochondria to cause rapid release of cytochrome c and AIF, producing apoptosis [41]. These responses were compromised when Akt was activated, or in chemoresistant cells [3]. We have demonstrated that in the presence of CDDP, P-p53 (Ser15) interacts with and promotes the trafficking of HKII from the mitochondria to the nucleus, downregulating its enzymatic activity and metabolism (glycolysis and OXPHOS). Although previous studies have shown that various chemical compounds (e.g., 2-DG and metformin) can induce mito-HKII detachment [42], we have demonstrated for the first time that activated P-p53 (Ser15) is required for intracellular HKII trafficking. In addition, whereas HKII localization in both the nucleus and cytoplasm has been reported, and HKII indeed possesses a nuclear localization signal [18,43], the specific mechanism of HKII nuclear localization has not been reported. Here, we have found that both p53 phosphorylation at Ser15 and the presence of NLS are required for the detachment of mito-HKII and its nuclear localization. Our study also suggests that Akt depletion and P-p53 (Ser15) activation synergistically facilitate nuclear localization, decreased protein content of HKII, and suppressed metabolism, inducing chemosensitivity. 

*TP53* mutation is prevalent in EOC, especially in the HGS histotype. Reles et al. reported that overexpression of p53 was found in 62% of HGS patients (110 of 178), and was associated with platinum resistance [44]. However, the level of p53 expression did not predict the clinical outcomes of patients [45]. Notably, our study revealed that nuclear HKII-P-p53 (Ser15) interaction and its downstream metabolic regulatory function of P-p53 (Ser15) are strongly associated with the clinical outcomes of EOC patients. We observed that P-p53 (Ser15) nuclear localization and nuclear HKII-P-p53 (Ser15) interaction were also evident in chemosensitive p53-mutant OV-2295 cells, and in clinical samples with p53 defects, suggesting that increased nuclear PLA signal could function as a potential biomarker in EOC with *TP53* mutations. 

Moreover, our studies implicate nuclear HKII-P-p53 (Ser15) in facilitating AIF-induced apoptosis. CDDP-induced, AIF-mediated apoptosis and AIF nuclear translocation from the mitochondria have been previously reported [27,34]. This study suggests that treatment with CDDP leads to the formation of a mitochondrial HKII–AIF–P-p53 (Ser 15) complex, whose recruitment to the nucleus facilitates AIF-induced apoptosis in chemosensitive EOC cells. These responses are, however, impaired in chemoresistant EOC cells harboring mutant *TP53*.

Established cell lines provide convenient and informative approaches for studying mechanisms of chemoresistance. However, their applicability in vivo must be established, since continuous passage may result in the loss of original tumor characteristics. To validate observations in EOC cell lines in vitro, we found that CDDP treatment significantly enhances the nuclear accumulation of HKII-P-p53 (Ser15) both in primary human EOC cultures/ascites and in tumor sections with platinum-sensitive cancers, defined by a PFI > 6 months. This signal was attenuated and restricted to mitochondrial regions in patients with platinum-resistant EOC. Regarding the prognosis of patients given adjuvant chemotherapy, the intracellular HKII-P-p53 (Ser15) interaction following platinum treatment may be predictive of the patient’s chemoresponsiveness and clinical outcome. In this study, primary human cancer cells and IHC sections were collected from EOC patients and treated primarily with carboplatin as a first-line standard chemotherapeutic regimen. Nuclear HKII-P-p53 (Ser15) interaction appears to be predictive of chemosensitivity in HGS EOC. Based on EOC cells and tumor section studies, this interaction may be relevant to other histological subtypes; this conclusion will depend upon evaluating larger numbers of patients with non-HGS histotypes. 

In this study, we have shown that nuclear HKII-P-p53 (Ser15) interaction functions as a prognostic biomarker by PLA. In clinical settings, this method has limitations compared to the detection of serum HK or immunohistochemistry; these limitations need to be overcome by streamlining the process. Compared with most serological or immunohistochemistry assay, which have focused on early- and late-stage diagnosis of ovarian cancer diagnosis, there have been few biomarkers for predicting prognosis and chemoresponsiveness. However, 70% of patients relapse with chemoresistance, and this often results in failure in treatment outcomes. Therefore, the development of prognostic biomarkers is required in order to satisfy unmet clinical needs for better treatment strategies and monitoring patient outcomes.

## 5. Conclusions

In conclusion, the present communication reports for the first time the importance of the intracellular trafficking of HKII and P-p53 (Ser15) in regulating glycolytic metabolism and chemosensitivity in EOC. Collectively, these results support the notion that p53 regulates nuclear HKII localization and plays a pivotal role in regulating cellular bioenergetics and chemosensitivity. Our translational findings that nuclear HKII-P-p53 (Ser15) interaction is implicated in chemosensitivity lend support to its clinical application as a promising biomarker for the monitoring of chemoresponsiveness in EOC patients during platinum-based therapy.

## Figures and Tables

**Figure 1 cancers-13-03399-f001:**
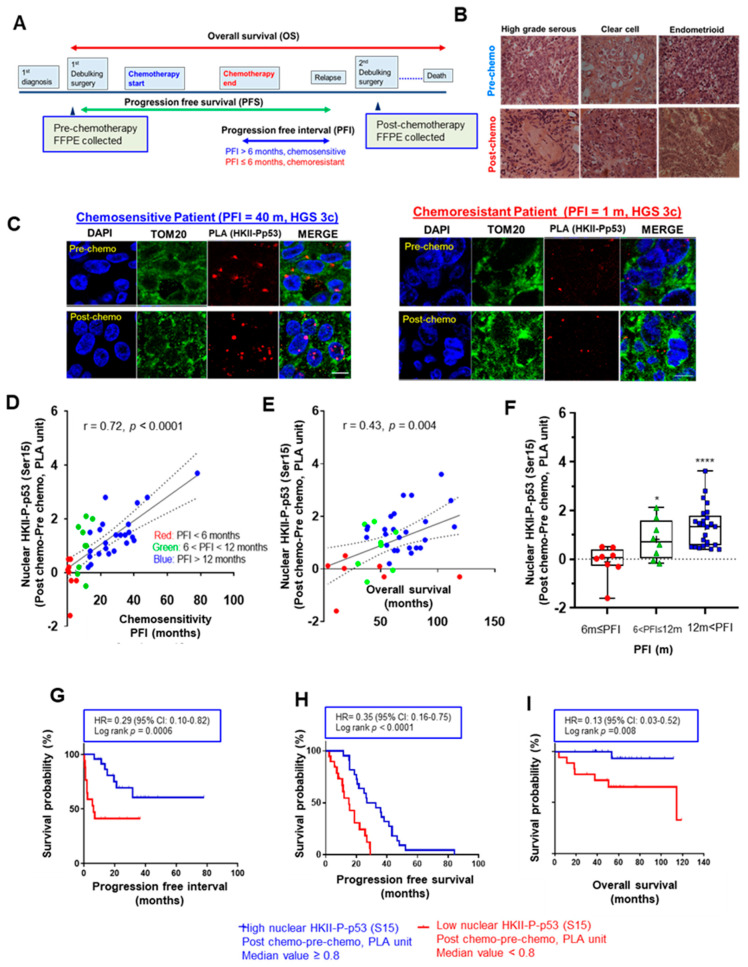
Chemotherapy-induced nuclear interaction of HKII-P-p53 (Ser15) in tumor tissue sections is associated with chemoresponsiveness in OVCA patients. (**A**) Paired pre- and post-chemotherapy high-grade serous ovarian tumor sections from the same patients (*n* = 41, total 82 sections) were collected at primary and secondary cytoreductive surgery, respectively. Samples were formalin-fixed and paraffin-embedded. Sections were stained with HKII and P-p53 (Ser15) antibodies. Lengths of progression-free survival (PFS, from cytoreductive surgery to recurrence of cancer), progression-free interval (PFI, from the end of first chemotherapy to recurrence of cancer), and overall survival (OS, from starting from cytoreductive surgery to death or last follow up) were calculated as pictured in the diagram. (**B**) Representative images of hematoxylin–eosin-stained sections of different histological subtypes of human EOC; scale bar: 10 μm. (**C**) In clinical ovarian cancer sections, the level of HKII-P-p53 (Ser15) interaction (proximity ligation assay (PLA) unit: red spot) and its cellular localization (blue: DAPI; green: TOM20) were assessed. Images depict PLA signals pre- and post-chemotherapy in IHC-stained sections of EOC collected from a chemosensitive patient (PFI = 40 m) and a chemoresistant patient (PFI = 1 m), respectively. (**D**) Significant positive correlation between the difference in PLA units for nuclear HKII-P-p53 (Ser15) interaction before and after chemotherapy, and the PFI of patients, in months (*n* = 41) (Difference in nuclear PLA units = HKII–P-PLA units for p53 (Ser15) interaction _(Post-chemotherapy)_–HKII-P-p53 (Ser15) interaction _(Pre-chemotherapy)_). (**E**) Significant positive correlation between the difference in PLA units before and after chemotherapy, and OS, in months (*n* = 40). (**F**) Box and whisker plots depict increased nuclear HKII-P-p53 interaction (PLA unit), and each box is divided according to a group of FFPE sections collected from patients with varied PFI as ≤ 6 m (red, *n* = 8), 6m < PFI ≤ 12 m (green, *n* = 8), and 12 m < PFI (blue, *n* = 25). Statistical significance was analyzed via the Wilcoxon test. (**G**–**I**) Higher nuclear PLA units for HKII-P-p53 (Ser15) interaction correlates with better PFI (**G**), PFS (**H**), and OS (**I**), as determined by Kaplan–Meier (KM) curves with hazard ratios (HRs). For PFI (**G**), the event was defined as chemoresistance (PFI ≤ 6 m; 1) whereas chemosensitive patients (PFI > 6 m; 0) were censored. For PFS, there were no censored patients. For OS, the event was death occurred in less than 60 m (OS ≤ 60 m) whereas chemosensitive patients (OS > 60 m; 0) were censored. The median value was assigned as the cutoff value (HKII-P-p53 (Ser15) interaction) to define low PLA expression with a score of < 0.8, and high PLA expression with a score of ≥ 0.8. The correlation was analyzed using the Pearson (r) method, and KM curves were stratified according to the logrank method. Censored patients were indicated on the KM curve as tick marks. For OS, the record of *n* = 1 is missing due to loss to follow-up. * *p* < 0.05 **** *p* < 0.0001.

**Figure 2 cancers-13-03399-f002:**
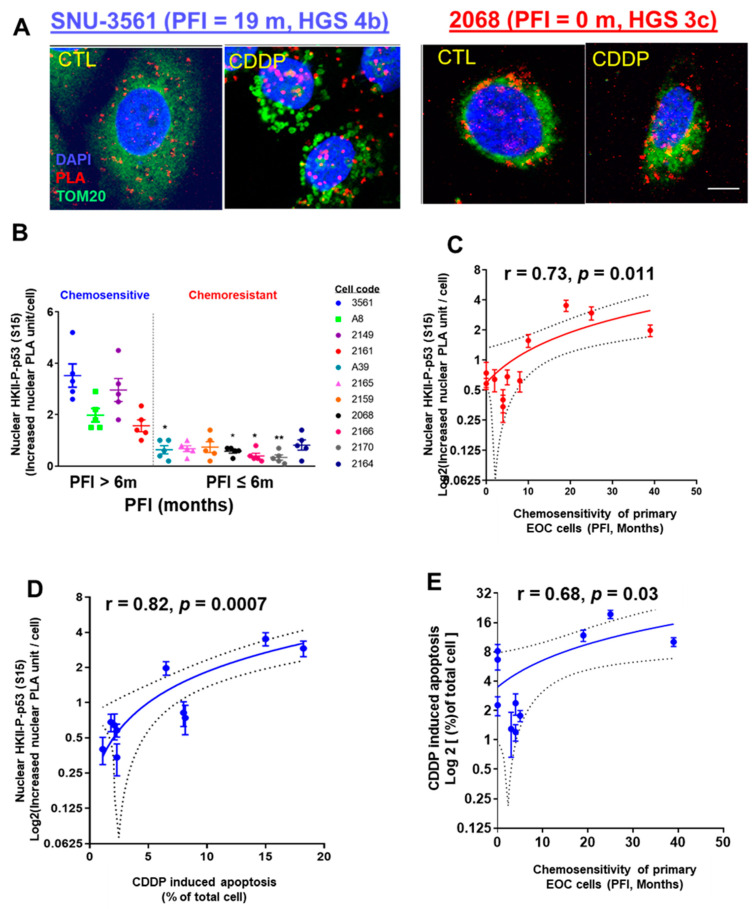
CDDP promotes interaction and intracellular trafficking of HKII-P-p53 (Ser15) to the nuclei of chemosensitive primary human OVCA cells. (**A**) Primary human EOC cell lines were established from chemosensitive SNU 3561 (PFI = 20 m) and chemoresistant 2068 (PFI = 0 m) cancers, and treated in culture with CDDP (10 µM, 24 h), or with DMSO as a vehicle control. Interaction of HKII and P-p53 (Ser15) (red dot) was measured via PLA. Cells were counterstained with anti-TOM20 (green) and DAPI (blue) before analysis with a Duolink Image Tool; scale bar: 10 μm. (**B**) Left panel: Eleven primary human EOC cell lines established from chemosensitive (PFI > 6 months, *n* = 4) or chemoresistant (PFI ≤ 6 m, *n* = 7) cancers were treated and analyzed as in (**A**) for a PLA interaction between nuclear HKII and P-p53 (Ser15). (**C**) Right panel: Correlation between the CDDP-driven increase in nuclear PLA units for the interaction of HKII and P-p53 (Ser15), and PFI. (**D**) Correlation between the CDDP-driven increase in nuclear PLA units for the interaction of HKII and P-p53 (Ser15), and CDDP-induced apoptosis measured by Hoechst staining, to judge chemosensitivity in cell culture. (**E**) Correlation between the CDDP-driven increase in nuclear PLA units for the interaction of HKII and P-p53 (Ser15), and PFI, to judge chemosensitivity in primary ovarian cancer cells. Error bars denote ± SEM (*n* = 5). Results are expressed as mean ± SEM (*n* = 5) (* *p* < 0.05, ** *p* < 0.01, CTL vs. CDDP). The correlations were analyzed using Pearson’s correlation (r) method.

**Figure 3 cancers-13-03399-f003:**
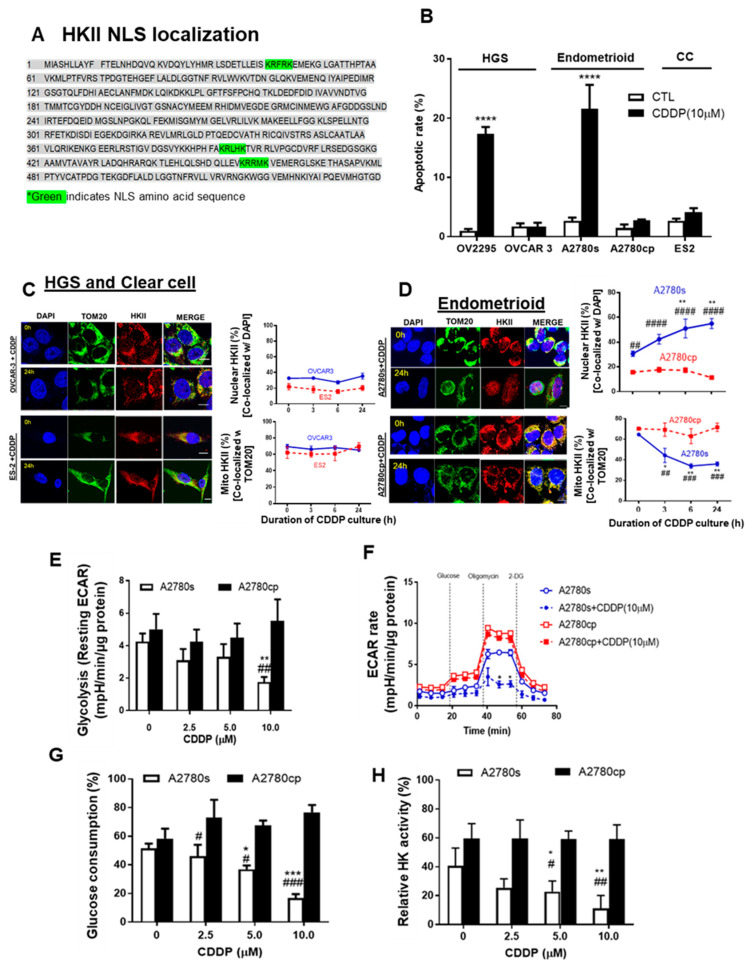
Mito-HKII is associated with chemoresistance and glycolysis in EOC. (**A**) Monopartite nuclear localization signal (NLS) present in the amino acid sequence of HKII (locus: XP_011531109). (**B**) High-grade serous (HGS; OV 2295, OVCAR-3), endometrioid (A2780s, A2780cp cells), and clear cell (CC: ES2 cells) lines were cultured with CDDP (0–10 µM; 0–24 h; DMSO as a vehicle), and apoptosis was measured using nuclear Hoechst staining. (**C**,**D**) EOC cell lines (OVCAR-3, ES2, A2780s, A2780cp) were treated as described in (**A**) and examined with confocal microscopy. Cellular localization of HKII (red; as % of total cells), TOM20 (green: mitochondrial marker), and DAPI (blue: nucleus marker) were analyzed using an Image Premier program. (**E**) A2780s and A2780cp cells were treated with CDDP (0–10 µM; 24 h; DMSO as a vehicle), and glycolysis (resting extracellular acidification rate (ECAR)) was measured. (**F**) ECAR was assessed over 72 min in cells treated after exposure to biomodulators (dashed vertical line). (**G**,**H**) A2780s and A2780cp cells were cultured as in (**G**), and glucose consumption (**C**; spent medium), and HK activity (**H**; cell lysates) were measured. Error bars denote mean ± SEM (*n* = 4). (* *p* < 0.05, ** *p* < 0.01, *** *p* < 0.001, **** *p* < 0.0001, CTL vs. CDDP; # *p* < 0.05, ## *p* < 0.01 ### *p* < 0.001, #### *p* < 0.0001, A2780s vs. A2780cp).

**Figure 4 cancers-13-03399-f004:**
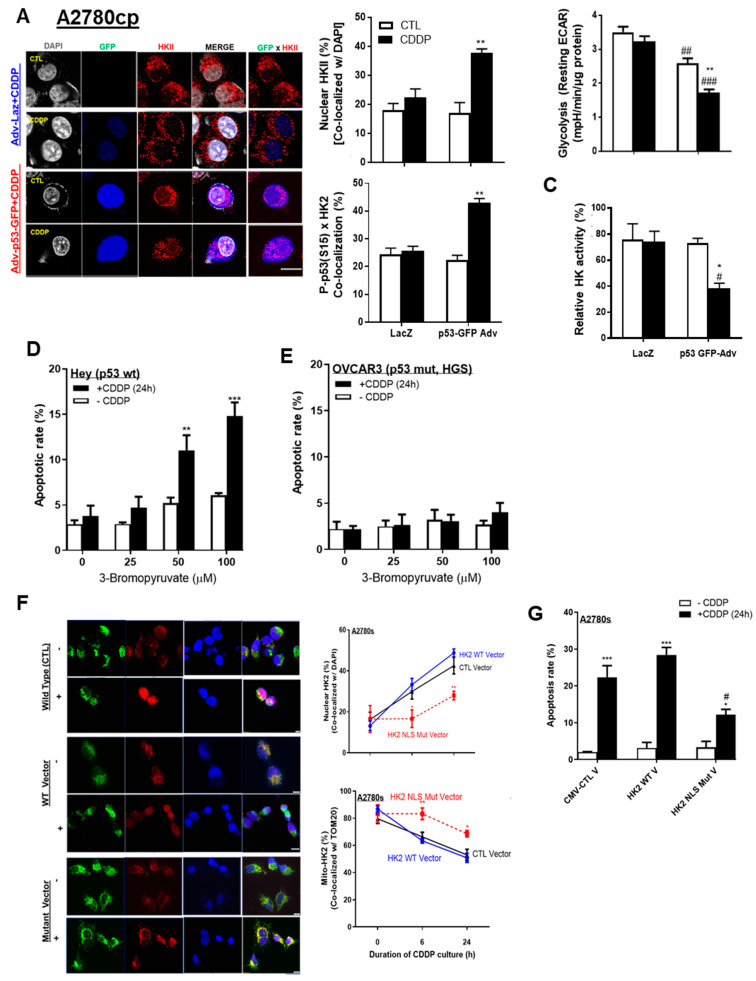
p53 and HKII NLS together are required for intracellular HKII trafficking. (**A**) A2780cp cells were transfected with either adenoviral construct (Adv)–LacZ (multiplicity of infection (MOI) = 1.0; 12 h), or Adv–p53 with GFP tag (MOI = 1.0; 12 h) and cultured with CDDP (0–10 µM; 24 h; DMSO as vehicle). Cellular localization of HKII (red; as % of total cells), P-p53 with GFP tag (blue), and DAPI (grey) was examined by confocal microscopy and quantified. (**B**) Glycolysis (resting ECAR) was measured using an XF96e Extracellular Flux Analyzer, while (**C**) HK activity was measured using cell lysate from A2780cp cells treated as described in (**C**). (**D**) Hey (p53 WT) and (**E**) OVCAR-3 (p53 WT) OVCA cells (HGS subtype) were pretreated with 3-bromopyruvic acid (3-BP; 6 h) and treated with CDDP (10 µM, 24 h) without changing the medium, followed by assessment of apoptosis with Hoechst nuclear staining. (**F**) A2780s cells were transfected with control CMV vector, HKII WT, and HKII NLS mutant vector (2.0 µg; 24 h). Transfected cells were seeded in 8-chamber slides and cultured with CDDP (0–10 µM; 24 h; DMSO as vehicle), followed by fixation for immunofluorescence (IF). Cellular localization of HKII (red; as % of total cells), TOM 20 (green), and DAPI (blue) was examined via IF microscopy (400×) and quantified. (**G**) Cells were treated as described and subjected apoptosis assessment. Scale bar: 10 μm. Error bars denote ± SEM (*n* = 4). (* *p* < 0.05, ** *p* < 0.01, *** *p* < 0.001, HKII CTL CMV vector vs. HKII NLS mutant vector) (Figure 4E: * *p* < 0.05, ** *p* < 0.01, CTL vs. CDDP; # *p* < 0.05, ## *p* < 0.01, ### *p* < 0.001, HKII CTL CMV vs. HKII NLS mutant). Scale bar: 10 μm.

**Figure 5 cancers-13-03399-f005:**
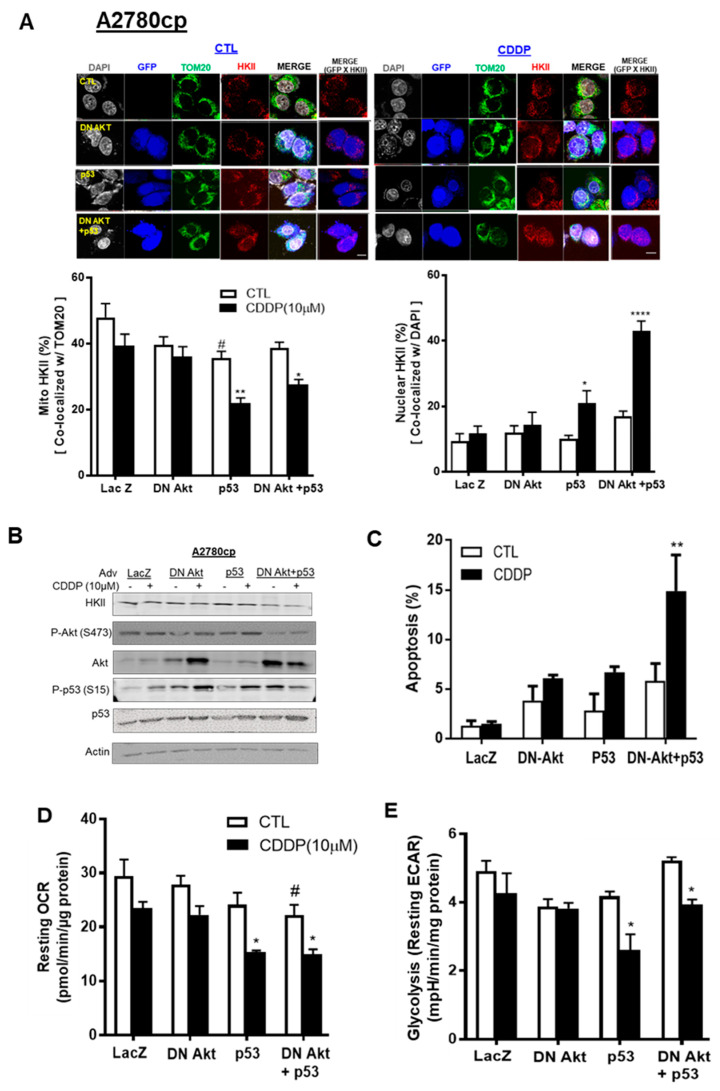
p53 reconstitution and Akt depletion together facilitate the detachment of HKII from the mitochondria. (**A**) A2780cp cells were transfected with either Adv–LacZ (MOI = 0.5 or 40; 12 h), Adv–p53 with GFP tag (MOI = 0.5; 12 h), or dominant negative (DN)-Akt Adv with GFP tag (MOI = 40), and cultured with CDDP (0–10 µM; 24 h; DMSO as vehicle). Cellular localization of HKII (red) as % of total cells, p53/P-p53 with GFP tag (blue), TOM20 (Green), and DAPI (grey) was measured using confocal microscopy and quantified. (**B**) A2780cp cells transfected as above were cultured with CDDP (10 µM; 24 h) followed by WB and (**C**) apoptosis measurement using Hoechst staining. (**D**) Oxygen consumption rate (OCR) and (**E**) glycolysis (resting ECAR) were measured in A2780cp cells infected with Adv–LacZ (MOI = 0.5 or 40.0; 24 h), Adv–p53 with GFP tag (MOI = 0.5; 12 h), or Adv–DN-Akt with GFP tag (MOI = 40; 12 h)—alone or together—seeded in 96-well plates, treated with CDDP (10 µM; 24 h), and analyzed using an XF96e Extracellular Flux Analyzer. Scale bar: 10 μm. Error bars denote ± SEM (*n* = 5). (* *p* < 0.05, ** *p* < 0.01, **** *p* < 0.0001, CTL vs. CDDP; # *p* < 0.05, Adv–LacZ vs. Adv–p53).

**Figure 6 cancers-13-03399-f006:**
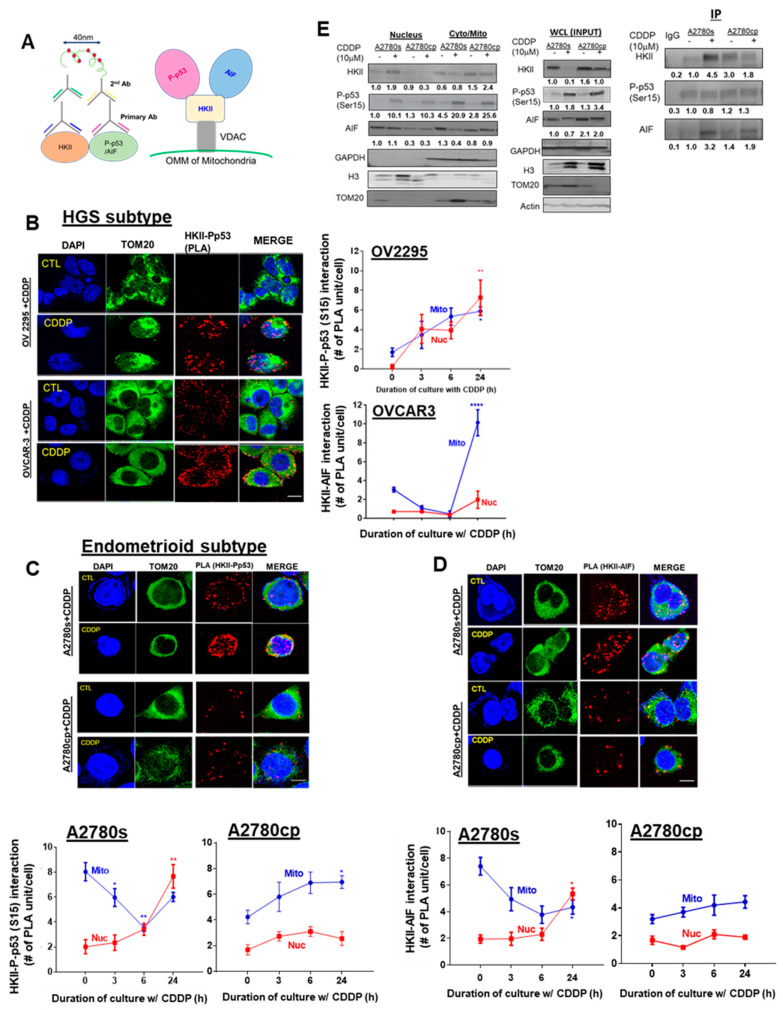
CDDP promotes interaction and intracellular trafficking of P-p53 (Ser 15)–HKII–AIF in the nucleus in chemosensitive OVCA cells. (**A**) Schematic diagram of the Duolink assay illustrating binding between HKII and P-p53 (Ser15)/AIF, and the molecular interaction (P-p53 (Ser15)–HKII–AIF) that was examined. (**B**,**C**) Examination of the nuclear interaction of the HKII-P-p53 (Ser15) PLA signal in different EOC subtypes. OV2295 and OVCAR-3 (HGS), and A2780s and A2780cp cells (endometrioid) were cultured with CDDP (10 µM; 24 h; DMSO as a vehicle). Interaction of (**D**) HKII–AIF in A2780s and A2780cp cells was assessed by using a Duolink Image Tool (Number of PLA units (A: red spot) with counter-staining of DAPI (blue), and TOM 20 (green). Error bars denote ± SEM (*n* = 5). (**E**) Right panel: A2780s and A2780cp cells were treated with CDDP (0–10 µM; 24 h). The nuclear fraction, cytoplasmic/mitochondrial combined fractions, and whole-cell lysate (WCL) were prepared. Protein contents of HKII, P-p53 (Ser15), AIF, TOM20 (mitochondrial loading control), histone H3 (nuclear loading control), GAPDH (cytoplasm loading control), and actin (loading control for WCL) were examined by WB. Left panel: A2780s and A2780cp cells were treated with CDDP (0–10 µM; 24 h). Nuclear fractions were collected and immunoprecipitated with anti-IgG (control) or anti-P-p53 (ser15). Protein–protein interaction was determined by IP/WB. P-p53 (Ser15) immunoprecipitates were immunoblotted (IP: anti-HKII; WB: anti-AIF). Results are expressed as mean ± SEM (*n* = 3) and were analyzed by two-way ANOVA and Bonferroni post-hoc test. (* *p* < 0.05, ** *p* < 0.01, **** *p* < 0.0001, *n* = 3).

## Data Availability

Relevant data supporting the findings of this study are available within the article and Appendix A and are available from the authors upon reasonable request.

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
