# Peer review of "Nuclear HKII–P-p53 (Ser15) Interaction is a Prognostic Biomarker for Chemoresponsiveness and Glycolytic Regulation in Epithelial Ovarian Cancer"

_cancers, 2021, doi:10.3390/cancers13143399_

Round 1

Reviewer 1 Report

Manuscript ID:   cancers- 1276270

In the new version of the article “Nuclear HKII-P-p53 (Ser15) interaction is a prognostic biomarker for chemo-responsiveness and glycolytic regulation in epithelial ovarian cancer” by Chae Young Han et al. cell culture conditions were described in details, including the number of passages of the primary EOC culture. ROC curves were described comprehensively.

Required minor correction:

Line 114:

In the Materials and Methods, the A2780 line is described as the endometrioid line; however, in the  reference 28 (doi:10.1038/ncomms3126) the exact histological origin of A2780 is not specified. This should be corrected in your manuscript. What are the IC50 values of cisplatin for A2780s and A2780cp cells in your experiments?

Double spaces should be corrected.

Reviewer 2 Report

The authors have properly addressed all the comments raised by the reviewer. The reviewer does not have any additional comments.

This manuscript is a resubmission of an earlier submission. The following is a list of the peer review reports and author responses from that submission.

Round 1

Reviewer 1 Report

In this article, the authors demonstrated a novel mechanism showing that activated phosphorylated-p53 (P-p53 Ser15) is required for the regulation of HKII intracellular trafficking and metabolic regulation in chemosensitive ovarian cancer but not in chemoresistant ovarian cancer harboring p53 mutation. They observed that increased nuclear HKII-P-p53 (Ser15) interaction is likely associated with chemosensitivity and better survival outcome in epithelial ovarian cell lines, human primary cells, and tumor sections. Nuclear HKII-P-p53 (Ser15) may function as a promising prognostic biomarker for chemoresponsivenss and metabolic dysregulation, enabling prediction of patients with poor prognosis for deciding better clinical strategy. The reviewer has some comments as below.

Major comments

  1. In Table S2, 2 cells are stated as “None” in chemo status column. It was unclear how did they quantify platinum sensitivity.
  2. They defined chemosensitivity by PFI, however, the PFI was censored in some patients according to the Kaplan-Meier curve (Figure 1G). It should be clarified how did they categorize censored patients.
  3. Related to the previous comments, since there are some censored patients, Figure 1D and 1E analysis are not appropriate way to analyze.
  4. It should be clarified how did they divide patients into high and low nuclear HKII-P-p53 in Figure 1G and 1H. Number at risk should be shown.
  5. Although they mentioned that carboplatin is standard chemo agent for EOC, they used CDDP in the study. It should be clarified why they used CDDP instead of carboplatin.
  6. From Figure S2, they stated that “Notably, strong HKII nuclear localization was observed in chemosensitive SNU-3561 (HGS, PFI = 16 m) even before CDDP treatment, suggesting that nuclear HKII is likely a determinant of chemosensitivity as basal level (Figure. S2)”. If they insist that base line nuclear HKII is chemosensitivity predictor, at least they should compare base line nuclear HKII among all used cell lines including primary culture.
  7. Figure 3C and 3D graph should be same format.
  8. From the ECAR results, they concluded that HKII nuclear localization decreased glycolysis. However, these cells are going to be apoptosis, thus, this kind of weaken cells have decreased metabolism. There are no convincing data it is due to HKII function but not weaken status.
  9. In Figure 4D, they suddenly brought up Hey cells. They should show the results of A2780s as p53 WT chemosensitive cells and OV2295 as p53 mutant chemo-sensitive cells.
  10. The reviewer does not understand DN Akt transduction part at all. The cells have endogenous Akt, what is rationale to ectopically express mutated Akt?
  11. In addition, western blot did not show increased p53 in p53 transduced cells (Figure 5B).
  12. In Figure 6E, cell fractionation did not work at all. There are less/no H3 band intensities in nucleus fractions, less TOM20 band intensities in Cyto/Mito fraction. In addition, there are huge difference of H3 intensity between with and without CDDP in whole cell lysate. In addition, input positive control is missing in IP blot.
  13. They stated in vivo experiment in the discussion section, but there were no in vivo experimental results.
  14. They wrote hypothesis in the conclusion section. Hypothesis should be introduction and proved things should be in the conclusion section.
  15. In the conclusion section, they stated that “In chemoresistant cells, treatment with CDDP fails to increase P-p53(Ser15), …” They should show the expression levels of P-p53 (Ser15) with and without platinum agent in all cell line used.
  16. Altogether, the reviewer agrees with that nuclear HKII-P-p53 (Ser15) in post chemotherapy is correlated with chemoresponsivenss and may be useful as a prognostic biomarker, but does not agree with that it may function as predictor of chemoresponsivenss and related with metabolic dysregulation.

Minor comments

  1. In the introduction, they wrote as “leading to poor overall survival (OS) rates (30-50%)”. It was not clear what is 30-50%. The reviewer assumes 5-year OS rate or something like that.
  2. In the next sentence, they wrote as “largely due to a defect in key tumor suppressors of apoptosis ( p53 ) or activation of oncogenes (Akt, PI3K)”. The reviewer dose not understand the meaning of “tumor suppressors of apoptosis”.
  3. They summarized their findings in the last paragraph of the introduction. It should move to discussion section.
  4. In the Material and Methods section, they mentioned about PA-1 cells, but it was in Table S1 and not used in this study.
  5. In Table S1, the subtype description should be consistent with the manuscript, e.g. use “HGS” instead of “Ovarian serous cyst adenocarcinoma”.
  6. In the Western blot section of Material and Methods, they wrote as “incubated with fluorescence-conjugated goat-anti-rabbit or anti-mouse secondary antibodies, …”, the reviewer believes that it is HRP-conjugated secondary antibody instead of fluorescent-conjugated one.
  7. Some of subtitle in the Material and methods are not appropriate. “Immunofluorescence microscopy” is a type of microscopy but not assay. It should be immunofluorescence staining or something like that. Adenoviral infection should be gene transduction by adenovirus.

Reviewer 2 Report

Thank you for the opportunity to review Nuclear HKII-P-p53 (Ser15) interaction is a prognostic biomarker for chemoresponsiveness and glycolytic regulation in epithelial ovarian cancer. I apologize for the late arrival of my comments, this is a large study that took time for me to work through. 

My recommendation is to accept this study in it's present form. This study has clearly been through multiple rounds of internal review between these authors and I'm having trouble finding any suggestions that might make this study better than it's current form. 

Minor comment: 

Lines 158-162, it would be helpful to add the catalog number for the antibodies, as this company provides several variations of these products. 

Lines 218-219. I find these formulas difficult to follow as text. I would find it easier to follow if these were broken out of the default text as formulas or as an image. 

Reviewer 3 Report

Manuscript ID:   cancers-1219770

The article “Nuclear HKII-P-p53 (Ser15) interaction is a prognostic biomarker for chemoresponsiveness and glycolytic regulation in epithelial ovarian cancer” by Chae Young Han et al. address an important issue of the HKII-P-p53 (Ser15) complex as a prognostic marker in ovarian cancer. The authors describe the phenomenon of nuclear co-localization of P-p53 and HK2 in a very interesting and methodically valuable manner. Moreover, the Authors highlight that the nuclear co-localization of HKII-P-p53 (Ser15) may be a promising biomarker in platinum-based therapy of EOC.

Some general remarks:

In the Introduction the references to the functional relationship of p53 and HK2 in EOC is missing (eg, no ref 38 published by the Authors). Provide them to refer to the Authors' previous research.

In the Materials and Methods the A2780 line is described as the endometrioid line; the appropriate reference should be cited as this feature is not described in the commercial documentation of A2780 cell line.

Detailed information about the composition of culture medium (e.g. antibiotics, hormones, insulin) and the number of passages should be provide in the description of the primary EOC culture.

Results: The sentence “Both ROC curves for PFI and OS showed AUC values > 0.5, 233 suggesting that nuclear HKII-P-p53(Ser15) has prognostic value for predicting chemosen-234 sitivity and survival outcome” is not entirely true. A poor ROC score for OS (<0.8) does not qualify for such a conclusion. The sentence should be revised. Moreover, the p value for the ROC analyses should be provided.

On the other hand, I have no critical remarks for the interpretation of the HR results, the correlation, or ANOVA. Figure S1: please provide the number of patients taken for the calculation.

Figure 2. OX axis descriptions are missing or they are cut-off. The Authors may consider to paste the OX axis as logarithmic in C-D figures for better visualization, as there is a large number of low-value points.

Figures 4 and 5. The GFP signal is always problematic when we want to locate it with another, but weaker signal. The DAPI, TOM20, and HKII signals are excellent, whereas GFP fluorescence is overexposed. The GFP signal might be numerically reduced and then merged. Image processing method can be described in the Materials and Methods for clarity. The clue of p53 and HKII co-localization is the superimposition of green and red images in Figure 5. Simultaneous merging of DAPI and TOM20 disturbs the reading. Please consider merging only green p53 and red HK2 while reduced green intensity, e.g. in a separate figure. Other photos are excellent.

Discussion: What is the advantage of the nuclear HKII-P-p53(Ser15) as a diagnostic  marker, over serum  HK levels or simple HK immunohistochemistry, or HK mRNA expression in EOC? This issue could be raised in the Discussion. Is it realistic to develop a diagnostic assay for HKII-P-p53(Ser15)?

Language: American and British English is confused - this needs to be corrected throughout the manuscript. Print errors (e.g. double spaces) and grammar errors should be corrected as well.